# Influence of Pre-Heat Treatment on the Deformation Behaviors, Microstructural Characteristics, and Mechanical Properties of a Continuously Cast Al-Cu-Mg Alloy during Continuous Extrusion Process

**DOI:** 10.3390/ma16083042

**Published:** 2023-04-12

**Authors:** Renbao Qin, Wentian Chen, Jie Tang, Fulin Jiang, Yonggang Chen, Hui Zhang

**Affiliations:** 1College of Materials Science and Engineering, Hunan University, Changsha 410082, China; 2State Key Laboratory of Powder Metallurgy, School of Materials Science and Engineering, Central South University, Changsha 410083, China; 3Hunan Province Engineering Research Center for High Performance Super Long Scale Light Alloy Materials, Yueyang 414005, China; 4Hunan Jinlianxing Special Materials Co., Ltd., Yueyang 414005, China

**Keywords:** Al-Cu-Mg alloy, heat treatment, deformation, microstructure, mechanical property

## Abstract

The presence of a second phase in Al-Cu-MG alloys, with various sizes and supersaturation-solid-solubility, which can be changed by pre-heat-treatment, could have remarkable influence on hot workability and mechanical performance. In the present work, a continuously cast 2024 Al alloy was homogenized and then subjected to hot compression and continuous extrusion (Conform) along with the initial as-cast alloy. The results showed that the 2024 Al alloy specimen with pre-heat treatment had a higher resistance to deformation and dynamic recovery (DRV) during hot compression process compared with the as-cast sample. Meanwhile, dynamic recrystallization (DRX) was advanced in the pre-heat-treated sample. After the Conform Process, the pre-heat-treated sample also attained better mechanical properties without additional solid solution treatment. The higher supersaturation solid solubility and dispersoids generated during pre-heat treatment was proved to play a key role in restricting boundary migration, tangling dislocation motion and promoting the precipitation of *S* phase, which raised resistance to DRV and plastic deformation and enhanced the mechanical properties.

## 1. Introduction

Al-Cu-Mg alloys have been extensively used as structural components in the aerospace and military industries due to their excellent comprehensive properties, such as high specific strength, superior high temperature strength, good formability and good fatigue resistance [1,2,3,4]. Numerous works have highlighted the dominant contribution of nano-size *S* precipitates and their precursors to the special mechanical performances [5,6,7]. For instance, Ringer et al. [8] concluded that peak hardness at 150 °C was associated with the formation of a independently nucleated *S* phase. Wang et al. [9] studied the aging hardening behavior of an Al-2.81Cu-1.05Mg-0.41Mn (wt.%) alloy at 150 and 190 °C, discovering that the formation of dense *S* precipitates predominated at the peak hardness stage. Kovarilk et al. [10] performed an artificial aging (AA) treatment at 180 °C for 24–96 h on an Al-Cu-Mg alloy and found that GPB zone was responsible for rapid hardening. Song et al. [11] carried out a two-stage aging treatment on 2024 Al alloy and proved the reliance of aging hardening on the precipitation of *S*’ particles. Esin et al. [12] obtained a precipitation map for AA2024-T3 alloy by conducting aging treatment at temperatures ranging from 85 to 250 °C, in which the maximum hardness occurred at the frontier between *S*-phase + GPB and *S*-phase regions.

In addition, coarse second phases, such as *T* phase (Al_20_Cu_2_Mn_3_), are frequently formed in Al-Cu-Mg alloys as well [13,14,15]. However, such coarse particles are of little benefit to mechanical properties [16], which can be reduced by homogenization treatment before hot processing. Previous studies [17,18,19,20,21] involving homogenization treatment mainly demonstrated the positive effects on mechanical properties of Al_3_Zr particles, rather than coarse second phases. The main influences can be concluded as follows: homogenization treatment can promote the precipitation of Al_3_Zr, which can inhibit recrystallization and the growth of recrystallized grains, thus refining the grain size during hot working and enhancing the mechanical properties. Second phase particles such as *T* phase were referred to play a similar role in improving resistance to recrystallization [22,23,24,25]. For example, Liu et al. [22] investigated the effect of *T* phase on the microstructure of a hot rolled Al-Cu-Mg alloy, and found that *T* phase could refine the grain size by inhibiting the growth of recrystallized grains. Li et al. [23] summarized the influences of *T* dispersoids as restraining the DRX and increasing the hot deformation activation energy during the hot compression process. In contrast, there has also been research suggesting that the particles could help to enhance the DRX process. Wang et al. [26] discovered that the Al_6_Mn phases formed during two-stage homogenization were useful for promoting the recrystallization process and inhibiting the growth of recrystallized grains. In addition, one other impact of second phase has been frequently mentioned, i.e., they could promote the formation of *S* precipitates, which could indirectly exert influence on alloys [16,22,27]. Nevertheless, the influences of coarse particles on the deformation behaviors during hot processing and the mechanical properties of final Al-Cu-Mg alloy products are still not well-investigated.

This paper aims to explore the influence of pre-heat treatment on the deformation behaviors, microstructural characteristics, and mechanical properties of a continuously cast Al-Cu-Mg alloy during continuous extrusion (Conform) process. Two initial pre-heat treatment states of 2024 Al alloy were designed before being subjected to deformation procedures, the continuous casted (CC) alloy without any treatment and the annealed continuous casted (ACC) alloy. Scanning electron microscopy (SEM), electron back scattering diffraction (EBSD) and transmission electron microscopy (TEM) were utilized to evaluate the microstructural evolution during the Conform Process and subsequent heat treatment. Hardness tests and uniaxial tensile tests were employed to examine the changes of mechanical properties.

## 2. Materials and Methods

Figure 1 shows the experimental procedures in this work. In the current work, two different initial state was prepared: one is CC alloy, which is the initial Al-4.4Cu-1.5Mg-0.6Mn-0.1Cr (wt.%, 2024 Al) alloy rod with a diameter of 16 mm, prepared using continuous casting; the other is ACC alloy, which is heat-treated at 350 °C for 6 h and 490 °C for 18 h before being quenched in water at room temperature (~20 °C). Subsequently, both the CC and ACC alloy specimens were submitted to deformation processes, including Conform and hot compression. Specimens after the Conform Process were subjected to heat treatment before carrying out characteristic tests, while specimens after hot compression was directly observed.

CC and ACC alloy were fed into the extrusion mold to produce the extruded rods with a diameter of 13 mm using the LJ400 (Riyue Machine Co., Ltd., Changzhou, China) continuous extrusion forming machine. The extrusion wheel speed was 4 rpm and the exit temperature for CC and ACC alloys were ~440 °C and ~470 °C, respectively. The processed alloys were quenched in water (~20 °C) immediately after extrusion.

Hot compression tests were also conducted on the CC and ACC alloy specimens to explore the influence of pre-heat treatment on the deformation behaviors of 2024 Al alloy under various temperatures and strain rates. The compression specimens were machined into cylindrical shapes with dimensions of 10 mm (diameter) × 15 mm (height). Isothermal hot compression tests were performed at temperatures of 300, 350, 400, and 450 °C and strain rates of 0.005, 0.05, 0.5, and 5 s^−1^, respectively, utilizing a computer controlled Gleeble-3500 thermomechanical simulator. The samples were resistance heated to the deformation temperature at a rate of 10 °C/s and held for 5 min to attain a stable and uniform temperature distribution throughout the sample. The samples were then deformed to a true strain of 0.7 at a constant temperature and strain rate. After compression, the specimens were immediately quenched in water (~20 °C).

To simulate the Conform Process, a three-dimensional elastoplastic commercial finite element package Deform-3D was carried out. The geometric dimensions of the die, groove roll and workpiece were identical to the experimental ones to assure the results were reliable. The isotropic hardening model based on the experimental hardening curves of the CC and ACC Al-Cu-Mg alloys was applied to the workpiece. The initial die temperature was assumed to be ~150 °C. Heat transfer coefficient between the die and workpiece was 30 N/sec·mm· °C. The Young’s modulus and Poisson’s ratio of the workpiece were 70,000 MPa and 0.33, respectively. The friction coefficient between the workpiece and groove wheel was defined as 0.95 and the lubricated friction coefficient on the die surface was set to 0.4 [28,29]. The extrusion die was considered a rigid body, while the workpiece was a rigid-plastic body meshed into 400,000 three-node grids.

After the Conform Process, heat treatment was implemented to examine the aging response of the extruded alloys. Two heat treatments were applied to both the extruded CC and ACC alloy specimens: (i) samples were directly subjected to AA treatment at 180 °C for various times ranging from 0 h to 36 h (T5); and (ii) samples were subjected to the same artificial aging treatment after solid solution (SS) treatment at 510 °C for 2 h followed by water (~20 °C) quenching (T6). Vickers microhardness measurements were performed by using a 0.2 kgf load for 15 s on a cross-sectional surface of the extruded and aged samples. For microhardness test, 5 points in different regions were tested, and the mean value and standard deviation was obtained to evaluate the hardness. Tensile specimens were machined parallel to extrusion direction (ED) at center according to the ASTM-E8. Uniaxial tensile tests were performed using a computer-controlled INSTRON3382 universal testing machine (Instron, Norwood, MA, USA) with a strain rate of 2 × 10^−3^ s^−1^. 3 samples were prepared for each test to guarantee the reliability of results.

To study the microstructure of 2024 alloys, specimens were collected from the rods parallel to extrusion direction, as shown in Figure 1. The samples were mechanically polished before being etched with Keller’s reagent and observed using CK-300 (Caikon, Shanghai, China) optical microscopy (OM). X-ray diffraction (XRD, Mini Flex 600, Mini-Flex, Ventura, CA, USA) from 5 to 90° (2θ) at a scanning rate of 10°/min using Cu-Kα radiation (λ = 1.5418Å) was used to identify the phase structure of the alloy. The polished specimens were also observed by TESCAN MIRA field emission scanning electron microscope (FSEM, TESCAN, Brno, Czech Republic). For EBSD analysis, the selected samples were ground and electropolished in an electrolyte consisting of 20% HClO_4_ and 80% C_2_H_5_OH under −25 °C for 25 s to remove the damage caused by grinding and mechanical polishing. Subsequently, EBSD measurements were performed on a JEOL-7900FSEM system (JEOL, Tokyo, Japan) equipped with an Oxford Instruments Nordlys Nano EBSD detector. The EBSD data was analyzed by orientation imaging microscopy version 6.2 (OIM 6.2) software. Samples for TEM test were cut from the center of heat-treated alloys. After mechanical thinning and double-jet electro polishing in a 30% nitric acid and 70% methanol solution at −20 °C, the samples finally became foils with a thickness of 50 μm and diameter of 3 mm. The TEM observations were performed using an FEI Tecnai G2 F20 TEM system (FEI, Inc., Valley City, ND, USA).

## 3. Results and Discussion

### 3.1. Microstructural Characteristics of the Initial CC and ACC 2024 Al Alloy Samples

The initial OM and SEM microstructures of CC and ACC alloy specimens are shown in Figure 2. As shown in Figure 2a, the OM microstructure of CC alloy displays as coarse grains made up of dendrites. The second phase can be seen to segregate along grain/subgrain boundaries before joining in a network in the SEM image (Figure 2b). According to the Energy Dispersive Spectroscopy (EDS) result, the main element segregated along boundaries are copper (Cu) accompanying with a small amount of magnesium (Mg). Cu and Mg are easy to enrich and form eutectic phases due to their low melting points. However, these reticular second phases will dissolve into Al matrix during subsequent homogenization treatment. Hence, the ACC alloy obtains higher supersaturation solid solubility. As seen in Figure 2c, plenty of dendrite structures are eliminated after homogenization, allowing for a much clearer view of grain boundaries. In Figure 2d, the segregation of second phase vanishes while some dispersoids are observed in the matrix after pre-heat treatment. According to the EDS analysis, the coarse particles retain lots of Cu elements, which should be the remnants of interdendritic segregation that were not dissolved. Meanwhile, in the finer particles, Manganese (Mn) was found, which implies these particles are newly formed *T* phases. These newly formed *T* phase particles along with solid solutes are the main differences between the CC and ACC alloy samples, which should be the key reason for their differences in subsequent deformation behaviors and mechanical properties. XRD results in Figure 3 basically correspond to the SEM observation. There is a Al_2_Cu phase along with Al_2_CuMg in the CC alloy, which constitute the main main composition of the interdendritic segregation. Meanwhile, these diffraction peaks disappear in the ACC alloy, which proves the vanishing of segregated phases. However, *T* phase is not recorded in the PDF cards which therefore could not be marked in the XRD result.

### 3.2. Deformation Behaviors of CC and ACC 2024 Al Alloy Specimens

#### 3.2.1. Flow Stress Evolution

Selected true stress-true strain curves obtained from isothermal compression tests are illustrated in Figure 4. The yield drop phenomenon is observed after the initial rapid increase in the flow stress, especially for the ACC alloys (Figure 4b,d). This phenomenon can be rationalized by dynamic deformation theories [30,31]. The mobile dislocation density suddenly multiplies during early stage, then dislocation motion and multiplication occurs at reduced stress levels [31,32]. Upon reaching the peak of the curves, the yield drop phenomenon occurs as the dislocations are unlocked by slipping over solute atoms and/or precipitates [33]. The ACC alloy obtain greater solid solubility during homogenization treatment, achieving higher strength during hot compression. The alloy elements may partly precipitate under the action of stress during hot compression, leading to the weakened in-solution strengthening. The flow stress level varies with different deformation temperature and strain rate. At the same deformation temperature, as shown in Figure 4a,b, the stress level increases with increasing strain rate. This can be summarized as positive strain rate sensitivity, while the stress is negatively dependent on deformation temperature. As illustrated in Figure 4c,d, as the temperature rises, the stress level gradually decreases owing to DRV and DRX induced by rising temperatures. In addition, the stress of the ACC samples (Figure 4b,d) is obviously greater than that of the CC samples (Figure 4a,c) under the same condition. In other words, homogenization prior to hot compression significantly improves the resistance to deformation for 2024 Al alloy.

#### 3.2.2. Constitutive Modeling

To evaluate the influence of deformation conditions and pre-heat treatment on the flow stress behavior of 2024 Al alloy, the Arrhenius constitutive equation and the Zener–Hollomon parameter in an exponent–type equation were employed [34,35]:(1)$$\dot{\varepsilon }=A\sigma^{n}\exp(-\frac{Q}{\mathrm{RT}})\text{, ασ < 0.8}$$
(2)$$\dot{\varepsilon }=\mathrm{Aexp}(\beta \sigma )\exp(-\frac{Q}{\mathrm{RT}})\text{, ασ > 1.2}$$
(3)$${\dot{\varepsilon }=A[\sinh(\alpha \sigma )]}^{n}\exp(-\frac{Q}{\mathrm{RT}})\text{, for all σ}$$
(4)$$Z=\dot{\varepsilon }\exp\left(\frac{Q}{\mathrm{RT}}\right){=A[\sinh(\alpha \sigma )]}^{n}$$
where $\dot{\varepsilon }$ is the strain rate (s^−1^), A is the structure factor (s^−1^), σ is the flow stress (MPa), n is the stress exponent, T is the absolute temperature (K), Q is the activation energy of hot deformation (KJ/mol), and R is the ideal gas constant (8.314 J·mol^−1^·K^−1^). A, n, α and β are material constants, and among them, α = β/n.

The Equation (4) can be rewritten as:(5)$${\left(\frac{Z}{A}\right)}^{\frac{1}{n}}=\sinh(\alpha \sigma )$$

Then, it is easy to obtain:(6)$$\sigma =\frac{1}{\alpha }\ln\left\{{\left(\frac{Z}{A}\right)}^{\frac{1}{n}}+\sqrt{{\left(\frac{Z}{A}\right)}^{\frac{2}{n}}+1}\right\}=\frac{1}{\alpha }\ln\left\{{\left(\frac{\dot{\varepsilon }\exp\left(\frac{Q}{\mathrm{RT}}\right)}{A}\right)}^{\frac{1}{n}}+\sqrt{{\left(\frac{\dot{\varepsilon }\exp\left(\frac{Q}{\mathrm{RT}}\right)}{A}\right)}^{\frac{2}{n}}+1}\right\}$$

The CC alloy is employed as an example to show the establishment procedures of constitutive equation. Taking the natural logarithms of both sides of Equations (1)–(3):(7)$$\ln\dot{\varepsilon }=\mathrm{lnA}-\frac{Q}{\mathrm{RT}}+\mathrm{nln}\sigma$$
(8)$$\ln\dot{\varepsilon }=\mathrm{lnA}-\frac{Q}{\mathrm{RT}}+\beta \sigma$$
(9)$$\frac{Q}{\mathrm{RT}}=\mathrm{lnA}-\ln\dot{\varepsilon }+\mathrm{nln}[\sinh(\alpha \sigma )]$$

The peak flow stress under different hot deformation conditions is substituted into Equations (7) and (8), as shown in Figure 5a,b. It is evident that the relationships $\ln\dot{\varepsilon }$–lnσ and $\ln\dot{\varepsilon }$–σ can be fitted by straight lines. Equal to the mean slopes of the lines in Figure 5a,b, the values of n_1_ and β are 8.7936 and 0.08854 MPa^−1^, respectively, and thus, α_1_ = 0.010069 MPa^−1^. The activation energy *Q* can be estimated by differentiating Equation (9):(10)$$Q=R\partial \ln\left[\sinh\left(\alpha \sigma \right)\right]/\partial \left(\frac{1}{T}\right){|}_{\dot{\varepsilon }}\cdot \partial \ln\dot{\varepsilon }/\partial {\ln[\sinh(\alpha \sigma )]|}_{T}=\text{RKn}$$

For a given strain rate, *K* is the slope of $\ln\left[\sinh\left(\alpha_{1}\sigma \right)\right]$−1/T line, and for a given temperature, n_2_ can be obtained by the slope of ln $\dot{\varepsilon }$−$\ln\left[\sinh\left(\alpha_{1}\sigma \right)\right]$ line. Then, a new value of α can be obtained, which can be substituted into Equation (10) again to get a newer value of K and n. The iterative process is repeated several times to guarantee an accurate result. The $\ln\left[\sinh\left(\alpha \sigma \right)\right]$−1/T and ln $\dot{\varepsilon }$−$\ln\left[\sinh\left(\alpha \sigma \right)\right]$ curves of the final iteration are plotted in Figure 5c,d. The final values of CC 2024 alloy are as follows: α_4_ = 0.019974, K_4_ = 6.6371, n_5_ = 4.0621, and thus, Q = 224.0402 KJ/mol. The value of A can be obtained from Equation (4) by taking the natural logarithms:(11)$$\text{lnZ = ln}\dot{\varepsilon }+\frac{Q}{\mathrm{RT}}=\mathrm{lnA}+n\ln\left[\sinh\left(\alpha \sigma \right)\right]$$

The values of lnZ at different deformation conditions can be calculated by substituting Q into the former equation in Equation (11). The linear relationship between Zener–Hollomon parameter lnZ and $\ln\left[\sinh\left(\alpha \sigma \right)\right]$ is plotted in Figure 5e, in which the intercept represents lnA. The value of A is 7.2710 × 10^14^. Figure 5f plots the lnZ − $\ln\left[\sinh\left(\alpha \sigma \right)\right]$ relationship of ACC alloy to give a better view of the comparison between CC and ACC alloy specimens. Apparently, Figure 5f owns a higher intercept than Figure 5e, which means a higher value of lnA and further a higher value of Q.

The modeling process of ACC alloy’s constitutive equation is similar to that of CC alloy. The values of material parameters A, n, α and Q in both CC and ACC alloy samples are listed in Table 1. Then, the constitutive equations of CC and ACC 2024 alloys are as follows:(12)$$\mathrm{CC}\;\mathrm{alloy}:\;\dot{\varepsilon }={7.2710\times 10}^{14}{[\sinh(0.019974\sigma )]}^{4.0621}\exp(\frac{224040.2}{8.31T})$$
(13)$$\mathrm{ACC}\;\mathrm{alloy}:\;\dot{\varepsilon }={3.2845\times 10}^{20}{[\sinh(0.014391\sigma )]}^{5.3905}\exp(\frac{297864.0}{8.31T})$$

The calculated values of peak stress can be easily obtained by substituting the deformation temperature and strain rate to Equations (12) and (13). An intuitive comparison of peak stress between calculated and tested values is given in Figure 6 to evaluate the accuracy of constitutive models. Sample numbers 1~4 were deformed under a temperature of 450 °C with various strain rates from 0.005 to 5 s^−1^, 5~8 were deformed under a temperature of 400 °C with various strain rates, and 9~12 and 13~16 were respectively deformed under temperature of 350 °C and 300 °C with different strain rates. It is clearly the case that the calculated and tested value of peak stress under same deformation conditions is very close, which indicates that the established constitutive model of peak stress of CC and ACC 2024 Al alloy can describe their rheological behaviors well.

#### 3.2.3. Microstructure of Hot Compressed CC and ACC 2024 Al Alloy Specimens

Figure 7 shows the EBSD inverse pole figure (IPF) and misorientation histograms of hot compressed CC and ACC 2024 Al alloy sample with a true strain of 0.7 at some selected conditions (temperature of 350 °C, strain rate of 0.5 s^−1^; 400 °C, 0.05 s^−1^; 450 °C, 0.005 s^−1^). The white lines were used to represent low angle boundaries (LAGBs, 2–15°) whereas the thick black lines were corresponded to high angle boundaries (HAGBs, >15°). The microstructure of hot deformed 2024 Al varies greatly with deformation temperature and strain rate as evidenced by different grain structure and boundary misorientation angle. For almost all selected conditions, the microstructure is mostly a recovered microstructure, i.e., large elongated grains containing substructure due to DRV along with occasional presence of a few fine equiaxed grains surrounded by HAGBs near/on the original grain boundaries. The former implies that DRV dominates the softening mechanism during hot compression. The latter indicates the occurrence of DRX and its preferential incidence near/on the boundaries. In addition, a color gradient between adjacent subgrains within the deformed original grains was observed, which indicates the occurrence of subgrain rotation inside the original grains. However, under these relatively high temperatures and low strain rates hot compression conditions, the dislocations annihilate each other and rearrange more thoroughly. With the progressive development of polygonization, nucleation of recrystallization is restricted. Thus, the absence of continuous dynamic recrystallization (CDRX) is caused by the lack of accumulated deformation energy to induce a considerable amount of recrystallization. DRV is enhanced while DRX becomes more absent with increasing temperature and decreasing strain rate, which can be proved by the increasing fraction of LAGBs (F_LAGBs_) in Figure 7a_2_–f_2_. Comparing the microstructure of ACC alloy with CC alloy under the same condition, two main differences can be found. The first one is an apparently coarser grain size which may inherit from the original coarse grain after homogenization but before deformation. The other is lower F_LAGBs_ and higher average misorientation angle, which may be related to the restriction of particles generated in homogenization on DRV and its dendritic structure being eliminated. To sum up, DRV progress is confined in ACC alloy. In fact, DRV involves diffusion or movement of atoms for dislocation annihilation and reordering through dislocation climb and cross-slip. As a high stacking fault energy metal, Al alloy should be easy to achieve DRV in. However, the solid solubility of ACC alloy maintains a high extent due to homogenization and water quenching. Therefore, these solutes, along with particles also generated in homogenization, can act as obstacles for atom diffusion and dislocation motion, such that DRV is confined.

### 3.3. Finite Element Analysis of Continuous Extrusion Forming

Figure 8 shows the distribution of deformation temperature, effective strain, mean stress, and effective strain rate of the CC and ACC alloys during the Conform Process along the transverse direction (TD) based on finite element analysis (FEM). The results of FEM uncover the significant influence of pre-heat treatment on the deformation behavior of 2024 Al alloy. As shown in Figure 8a–c, temperature gradually increases after being fed into cavities as a result of severe friction and plastic deformation. Upon reaching the exits of models, the highest temperature is obtained. The ACC alloy owns higher outlet temperature (471 °C), which is 29 °C higher than that of CC alloy. The activation energy of hot deformation of ACC alloy in Table 1 is obviously higher than CC alloy, which means higher resistance to hot deformation of the former than the later. As a result, the outlet temperature of ACC alloy is higher than the CC alloy. Temperature at the exit of the model is of great benefit to help achieving a greater online solid solution. Meanwhile, due to the uneven friction effects, the edge temperature around the die exit is slightly higher than the center region, as shown in the point tracking curves in Figure 8c. The effective strain in the cavities of the CC and ACC alloys is at a close level, as shown in Figure 8d–f. The equivalent strain increases substantially after the intense shear zone and finally results in high strain accumulation around the die exit. The corresponding point tracking curves (Figure 8f) uncovers the uneven strain distribution between central and edge region, i.e., the effective strain in the edge region is much higher than that of the central region. In Figure 8g,h, the mean hydrostatic stress is maintained at a high level in Conform cavities, which is beneficial for eliminating defects such as micro-cracks and for enhancing the quality of products. And the mean stress of ACC alloy in cavity is apparently higher than CC alloy. Further, the effective strain rate of different alloys is also quite different (Figure 8i,j), and higher strain rate in the ACC alloy will bring it additional shear deformation. Higher exit temperature and hydrostatic stress suggest higher deformation resistance which may rationalize the considerable effect on deformation behavior and microstructure caused by pre-heat treatment.

### 3.4. Microstructure Characterization after the Conform Process

The EBSD microstructures of the CC and ACC 2024 Al alloys after the Conform Process and following SS treatment are illustrated in Figure 9. There are both elongated coarse grains and fine recrystallized grains in the CC alloy after the Conform Process, as shown in Figure 9a_1_. Numerous LAGBs still exist in the extruded alloy as the average misorientation angle (θ_AV_) and fraction of high angle grain boundaries (F_HAGBs_) of the CC specimen are only 23.20° and 56.36%, respectively. The results imply that DRX in this alloy is not completely evolved. The misorientation statistics rise up to 27.3° and 73.89% after SS treatment as shown in Figure 9b_2_. The extruded ACC alloy, on the other hand, exhibits coarser grains and fewer substructures in the IPF map (Figure 9c_1_), suggesting a more adequate DRX may be achieved. Higher θ_AV_ and F_HAGBs_ in Figure 7c_2_ are in favor of this point. As seen in Figure 9d_1_,d_2_, the grain morphology after SS treatment is slightly coarser and the F_HAGBs_ is slightly higher than the extruded ACC alloy. The θ_AV_ and F_HAGBs_ of the SS treated ACC alloy (ACC-SS) is also higher than the SS treated CC alloy (CC-SS), which further proves the enhanced effect of pre-heat treatment on recrystallization. It was mentioned in Section 3.2.3 that solutes and particles created during homogenization could hinder dislocation motion, which prevented dislocation from annihilation. As a result, more dislocations are retained in the matrix and more energy is accumulated, advancing the nucleation of recrystallization.

### 3.5. Mechanical Properties

To investigate the aging hardening response, both the as-extruded and SS treated specimens were subjected to an AA treatment at 180 °C for 36 h. Figure 10a shows the microhardness evolutions with aging time of the CC and ACC specimens. The CC sample directly subjected to AA (CC-T5) is insensitive to AA treatment where a slight increase in hardness is observed. Meanwhile. the ACC sample benefits a lot from T5 treatment, whose peak hardness is close to the CC and ACC samples after SS and AA (CC-T6 and ACC-T6). Tensile tests were performed on samples that had been aged for 16 h in order to unify the variation of aging time. The results are shown in Figure 10b and Table 2. The variation trend of tensile strength generally corresponds to hardness results. The strength of CC-T5 specimen is the lowest, as expected, while the CC-T6 sample shows higher strength but lower elongation. The strengthening may benefit from good solution extent caused by SS treatment, which enhances the aging response. The tensile strength of the directly AA treated ACC (ACC-T5) specimen is 481 MPa, which is 43.6% higher than the CC-T5 specimen. Tensile strength and elongation of the ACC sample after T6 tempering are 485 MPa and 18.1%, respectively, which are 13.6% and 85.1% higher than the CC-T6 sample. In conclusion, pre-heat treatment significantly enhances the comprehensive mechanical properties of 2024 Al alloy. On the other hand, the mechanical performance of the ACC-T5 sample is substantially better than that of the CC alloy and is comparable to the ACC-T6 sample. In other words, the ACC alloy achieves a great online solution after the Conform Process, which provides a novel routing to obtain good performance.

### 3.6. TEM Microstructure Characteristics

The TEM micrographs of selected 2024 Al alloys subjected to various pre-heat treatments and AA procedures are shown in Figure 11. As shown in Figure 11a, the CC-T5 sample exhibits relatively clear and straight subgrain boundaries. Particles including coarse ones generated during homogenization and fine precipitates generated during AA treatment are less distributed in the matrix in Figure 11b. The selected area electron diffraction (SEAD) pattern also demonstrates the absence of precipitates. The lack of precipitates could be attributed to the low solid solubility. On the one hand, the segregation in CC alloy is rich, which means a low initial solution extent. On the other hand, the outlet temperature after Conform (~440 °C) is insufficient to introduce an online solution. Dislocations are also uncommon in this alloy, which can be connected to the previously mentioned DRV in this high stacking fault energy metal. The absence of precipitates and dislocations should be responsible for the low strength of the CC-T5 alloy. There are also few dislocations in the CC-T6 specimen, similar to the CC-T5 specimen, as shown in Figure 11c,d, accompanying with the presence of dense *S* precipitates.

As is known, *S* phase has been reported to precipitate with the {120}_Al_ habit planes and grow along {100}_Al_ direction. Considering the orientation relationships between *S* phase and the Al matrix are [100]_Al_//[100]*_S_*, [02$\bar{1}$]_Al_//[010]S, [012]Al//[001]*_S_*, there are 12 equivalent variants of *S* phase to orientation relationship [12,36,37,38]. *S* precipitates have two morphologies in Figure 11d. The first is an ordinary needle-like *S* phase with an average length of around 460 nm, which is marked in a yellow rectangle. Furthermore, this kind of *S* precipitates are perpendicular with each other, and they should be recognized as the *S* variants along (0$\bar{2}$1)_Al_//(001)*_S_* and (012)_Al_//(001)*_S_* [39]. The second is distinctive, which looks like intermittent lines connected along <0$\bar{1}$3>_Al_ direction and is marked in a red rectangle. In fact, these fine dots instead of lines should be recognized as *S* phases according to previous works [36,38,39,40,41,42,43], and these intermittent points should be considered as the cross-sections of rod-like *S* precipitates. In other words, these rod-like *S* phases are parallel to the observation direction <001>_Al_ and arranged along <0$\bar{1}$3>_Al_ direction. A precipitation hardening mechanism is objectively present and functioning in this alloy. Compared to the CC samples, the ACC samples (both ACC-T5 and ACC-T6) have more considerable dislocations and precipitates, as shown in Figure 11e–h. The average length of *S* precipitates in Figure 11f is about 197 nm, which is much finer than that in Figure 11h (~623 nm). Nonetheless, the interaction between *S* precipitates and dislocations in both ACC-T5 and ACC-T6 alloy apparently dominates the enhancement mechanism of the ACC alloy.

Another distinct phenomenon is the even distribution of dispersoids produced during homogenization. Additional observation on these particles is carried out for the ACC alloy. As shown in Figure 12, numerous dislocation helices are trapped by dispersoids in the matrix and lots of *S* precipitates are observed around dispersoids and dislocations. The particles can act as obstacles for dislocation motion, resulting in the accumulation of dislocations and energy. Thus, they are also the preferred location for precipitation. This mechanism is believed to help enhance the mechanical properties of the ACC alloy. Figure 12c shows the EDX analysis result of the dispersoid marked by red star in Figure 12b. The main alloy elements are Cu and Mn, which indicates that the dispersoids observed are *T* phases and they are original from the pre-heat treatment.

### 3.7. Discussion

Pre-heat treatment significantly affected the deformation behaviors, microstructures, and mechanical properties of 2024 Al alloy. Firstly, the higher flow stress level in Figure 4 and mean stress in Figure 8 of the ACC alloy revealed that the increasing deformation resistance occurred after pre-heat treatment. The main plastic deformation mechanisms in Al alloy are grain boundary sliding (GBS) and dislocation glide (DG) [44]. The former dominates under low strain rate condition or at early deformation stage while the latter works under high strain rate conditions or at a later stage [45]. As a result, the deformation resistance can be influenced by any factors that exert on GBS and DG. The particles generated during pre-heat treatment could easily inhibit grain boundary migration and dislocation motion, causing difficulty in deformation. In addition, an increase in the solutes content after homogenization could play similar role in increasing the alloy’s deformation resistance. Previous studies [46,47] have reported the increasing deformation resistance of Al-Mg-Si alloy induced by increasing Mg-Si solution extent.

Secondly, the DRV was restrained while the DRX was advanced for the ACC alloy during deformation process, as proved by the deformed microstructures in Figure 7 and Figure 9. During the hot compression process, dislocations proliferated rapidly with the accumulation of strain, resulting in the formation of dislocation tangles and cells. However, the high temperature provided thermal activation condition for the DRV. The climbing of edge dislocations, cross slip of screw dislocations and unpinning of the dislocation node took place in the CC alloy to decrease the dislocation density. However, the dispersoids and solution atoms in the ACC alloy restricted dislocation motion, and thus confined the progress of DRV.

There are two different focuses regarding how the particles affected the DRX process: (i) they can enhance the DRX mechanism though increasing the drive force and/or stimulating the nucleation [26,48], and (ii) more researchers think it can increase the recrystallization resistance by increasing the Zener drag force on dislocations and subgrain boundaries [17,18,19,20,49]. The stored energy can be expressed as follows [50]:(14)$$P_{D}{=\alpha \mathrm{Gb}}^{2}\rho$$
where γ is a constant equal to 0.5, and ρ, G, and b are the dislocation density, shear modulus, and Burgers vector, respectively. While the Zener drag force can be expressed as [51]:(15)$$P_{Z}=3\gamma \frac{F_{V}}{d}$$
where γ is grain boundary energy, and $F_{V}$ and d are the volume fraction and mean size of the particles, respectively.

According to Equation (14), the stored energy mainly depends on the dislocation density, which is affected by deformation conditions such as strain rate, deformation temperature and second phase distribution. On account of the same process parameters adopted in the Conform Process, the stored energy level mainly depends on the distribution of the second-phase particles. Thus, the dislocation density can be further described as [52]:(16)$${\rho =\rho }_{s}+\Delta {\rho =\rho }_{s}+6\frac{s}{b}\frac{F_{V}}{d}$$
where $\rho_{s}$ is the dislocation density generated in the matrix, Δρ is the increment of dislocation density induced by the particles, and s is shear strain. The ACC alloy obtained many fine particles after homogenization (Figure 2) and retained them after Conform process (Figure 11 and Figure 12). The presence of these particles helped prevent the dislocations from disappear during the DRV, and the results in Figure 11 proved that the ACC alloy had a higher dislocation density than the CC alloy. Combining Equations (14) and (15), the driving force is expressed as:(17)$${P=P}_{D}{-P}_{Z}={0.5\mathrm{Gb}}^{2}\rho_{s}+3(\mathrm{sGb}-\gamma )\frac{F_{V}}{d}$$

Herein, the parameters were determined to be [53,54,55]: G = 18 GPa, b = 0.286 nm, γ=0.32 J·m^−2^. As for $s=\sqrt{3}\varepsilon_{\mathrm{Conform}}$, according to von Mises, the equivalent strain $\varepsilon_{\mathrm{Conform}}$ was determined to be approximately 3 by FEM in Figure 8f. Thus, the value of (sGb-γ) is apparently > 0, which means the secondary particles bring a higher driving force. Thus, the higher driving force of the ACC alloy compared to the CC alloy is verified to be related to the existence of the second phase particles in the former and their lack in the latter. To further understand how the particles enhance the DRX process, a schematic illustration is shown in Figure 13. The stored energy increased as the dislocations accumulated because of being pinned by the particles. As it can provide adequate driving force, the region near the particles can act as the preferred site for DRX to form new recrystallized grains. After recrystallization, the stored energy is consumed, and its consumption is represented by a reduction in shade.

Thirdly, pre-heat treatment helped to achieve enhanced mechanical performances after the Conform Process, as shown in Figure 10. It is known that AA treatment is regarded as the most effective method to strengthen 2024 Al alloy by inducing nano precipitates. The dominant strengthening precipitates in 2024 Al alloy is *S* phase. As it is not coherent with the matrix, *S* phase is difficult to directly precipitate. The generally precipitation sequence in Al-Cu-Mg alloys is as follows: supersaturated solid solution (SSS) → Cu-Mg cluster → GP → *S*’ → *S* [10,38,40,56,57]. The high-density dislocations from plastic deformation often act as preferential nucleation for *S* precipitates [41,43,49,58]. It is worth noting that in Figure 11e–h and Figure 12, the matrix of the ACC samples exhibit some coarse rod-like phases, which are generated during long-time homogenization, as mentioned in Figure 2d. Being incoherent with the Al matrix, these phases can capture dislocations (Figure 12); hence, the interface between matrix and the secondary phase provides advantageous nucleation positions for *S* phase [16,59,60]. As a result, large number of precipitates and dislocations were found to form around the particles in the ACC alloy. Based on the above analysis, the dispersoids trapped dislocations and promoted the production of *S* precipitates, finally leading to the strengthening of alloys. Another possible explanation for pre-heat treatment’s beneficial influence on mechanical properties is the good initial supersaturated solid solution caused by the water quenching following homogenization. As the ACC alloy has a high outlet temperature (~470 °C) after the Conform Process, which is approaching the SS treatment temperature of 2024 Al alloy. Thus, the ACC alloy achieves a better online solution than the CC alloy. As a good supersaturated solution has been achieved in the ACC alloy, the ACC alloy directly subjected to AA treatment after extrusion can obtain good mechanical properties as well. As shown in Figure 10, the ACC sample after T5 treatment presented equivalent strong performances to the T6 treated sample. This new process, without solid solution treatment, not only shortened the manufacturing process and improved production efficiency, but also saved on production cost.

## 4. Conclusions

In this work, hot compression and the Conform Process were conducted on both CC and ACC alloys to investigate the influences of pre-heat treatment on deformation behavior, microstructural evolution and mechanical properties of 2024 Al alloy. The major conclusions could be summarized as follows:(1)The CC and ACC samples showed different flow behavior during hot compression, owing to the variant deformation mechanism. ACC samples after pre-heat treatment owned higher resistance to deformation, which reflected higher flow stress and higher value of *Q*. DRV happened in both CC and ACC samples, while DRX was advanced in ACC sample during hot deformation.(2)After the Conform Process, ACC alloy attained the enhanced online solid solution, which could be directly subjected to AA treatment to achieve good mechanical performance. The tensile strength of ACC-T5 sample (481 MPa) was identical to ACC-T6 (485 MPa) with good ductility, which exceeded the mechanical properties of CC-T5 and CC-T6 samples remarkably.(3)After pre-heat treatment, the higher supersaturation solid solubility and numerous dispersoids (*T* phases) were observed in the ACC sample by dissolving the coarse particles in CC specimen. TEM microstructures showed that large number of dislocations were captured by the particles and many *S* phases precipitated around the particles. Tangling dislocations and stimulating precipitation of *S* phase were the main reasons for the enhanced mechanical properties.

## Figures and Tables

**Figure 1 materials-16-03042-f001:**
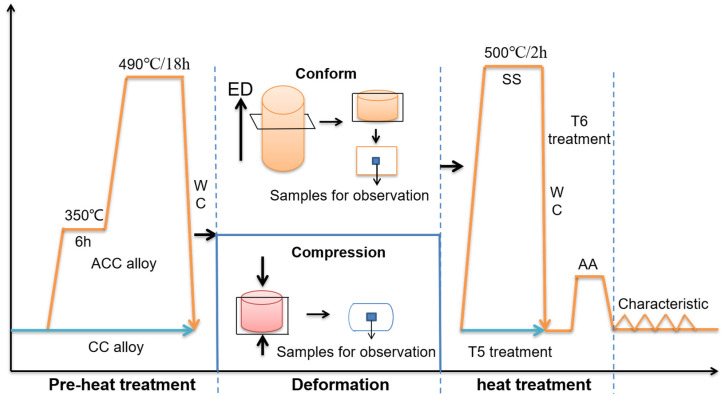
Schematic diagram of the experimental procedures.

**Figure 2 materials-16-03042-f002:**
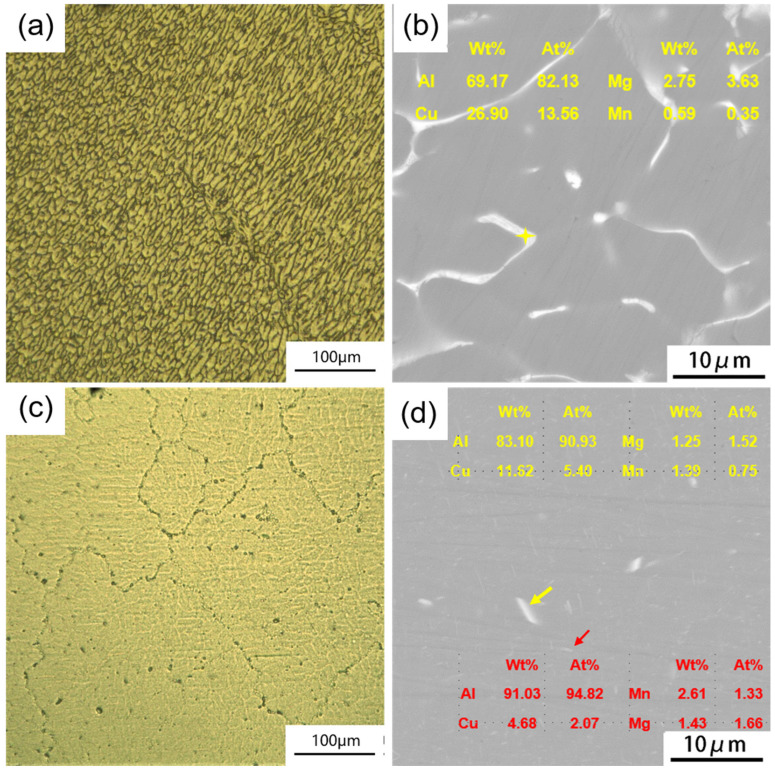
Initial OM and SEM microstructures of 2024 Al alloy: (**a**,**b**) CC alloy specimen and (**c**,**d**) ACC alloy specimen.

**Figure 3 materials-16-03042-f003:**
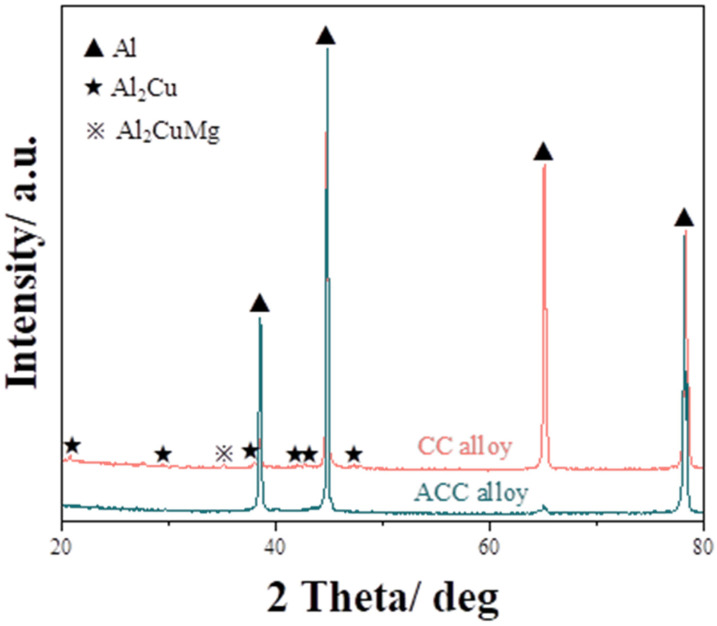
X-ray diffraction patterns of the CC and ACC alloy.

**Figure 4 materials-16-03042-f004:**
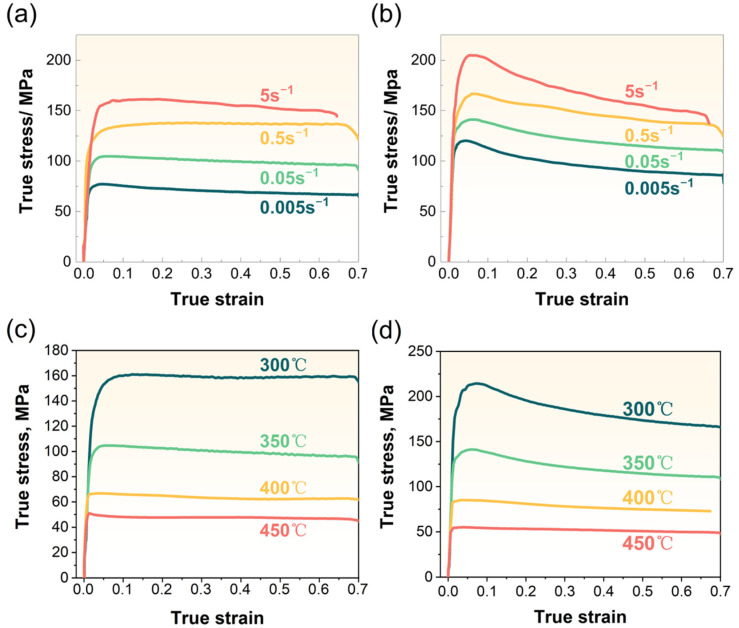
True stress-strain curves deformed under 350 °C at different strain rates of (**a**) CC samples, (**b**) ACC samples, and deformed at 0.05 s^−1^ for various temperatures of (**c**) CC samples, (**d**) ACC samples.

**Figure 5 materials-16-03042-f005:**
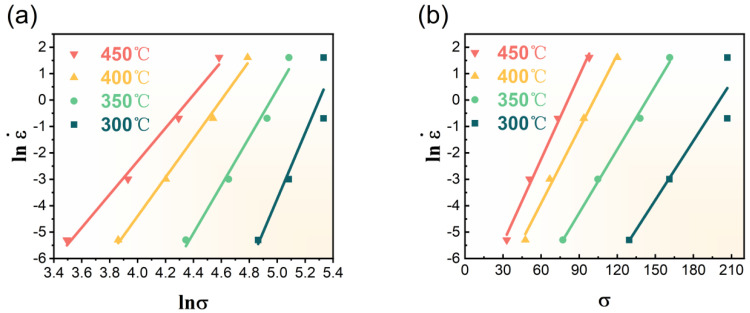
(**a**–**e**) Linear relationship fits of CC alloy: (**a**) $\ln\dot{\varepsilon }$–lnσ; (**b**) $\ln\dot{\varepsilon }$–σ; (**c**) $\ln\left[\sinh\left(\alpha_{1}\sigma \right)\right]$–1/T; (**d**) ln $\dot{\varepsilon }–\ln\left[\sinh\left(\alpha_{1}\sigma \right)\right]$; (**e**) lnZ–$\ln\left[\sinh\left(\alpha \sigma \right)\right]$; (**f**) linear relationship between lnZ–$\ln\left[\sinh\left(\alpha \sigma \right)\right]$ of ACC alloy.

**Figure 6 materials-16-03042-f006:**
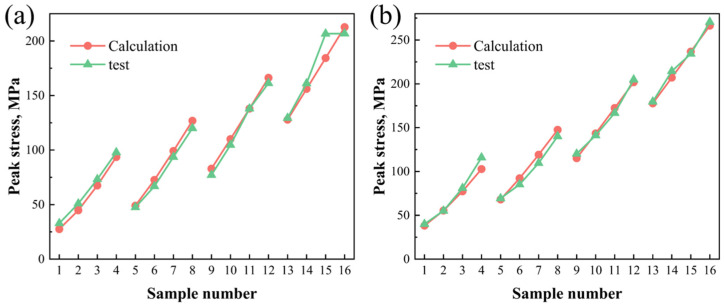
Comparison of calculated and tested peak stress under different deformation conditions: (**a**) CC alloy and (**b**) ACC alloy.

**Figure 7 materials-16-03042-f007:**
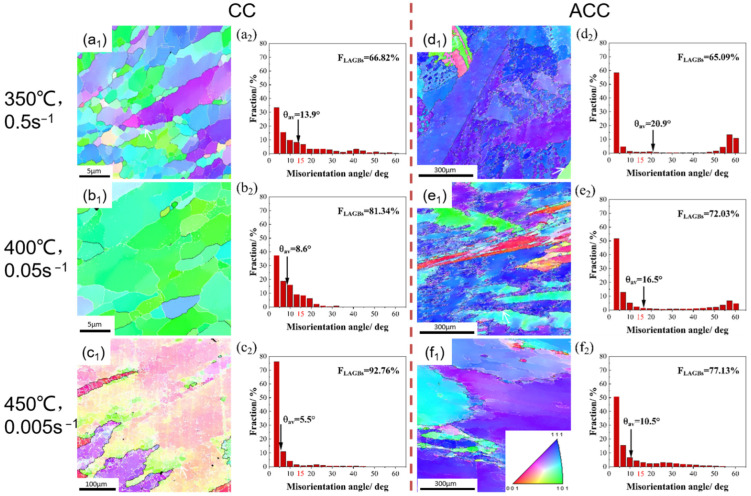
EBSD IPF maps and misorientation angle distribution images of hot compressed CC and ACC 2024 Al alloy specimens at strain of 0.7. (**a_1_**) IPF map of CC alloy deformed under 350 °C, 0.5 s^−1^; (**a_2_**) misorientation angle distribution of CC alloy deformed under 350 °C, 0.5 s^−1^; (**b_1_**) IPF map of CC alloy deformed under 400 °C, 0.05 s^−1^; (**b_2_**) misorientation angle distribution of CC alloy deformed under 400 °C, 0.05 s^−1^; (**c_1_**) IPF map of CC alloy deformed under 450 °C, 0.005 s^−1^; (**c_2_**) misorientation angle distribution of CC alloy deformed under 450 °C, 0.005 s^−1^; (**d_1_**) IPF map of ACC alloy deformed under 350 °C, 0.5 s^−1^; (**d_2_**) misorientation angle distribution of ACC alloy deformed under 350 °C, 0.5 s^−1^; (**e_1_**) IPF map of ACC alloy deformed under 400 °C, 0.05 s^−1^; (**e_2_**) misorientation angle distribution of ACC alloy deformed under 400 °C, 0.05 s^−1^; (**f_1_**) IPF map of ACC alloy deformed under 450 °C, 0.005 s^−1^; (**f_2_**) misorientation angle distribution of ACC alloy deformed under 450 °C, 0.005 s^−1^.

**Figure 8 materials-16-03042-f008:**
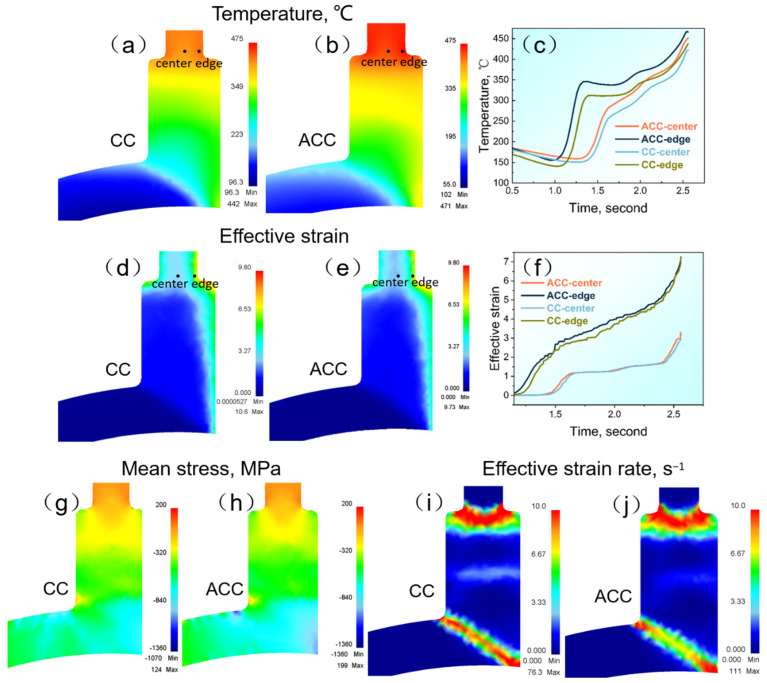
Finite element analysis during the Conform Process: (**a**–**c**) deformation temperature; (**d**–**f**) effective strain; (**g**,**h**) mean stress; (**i**,**j**) effective strain rate.

**Figure 9 materials-16-03042-f009:**
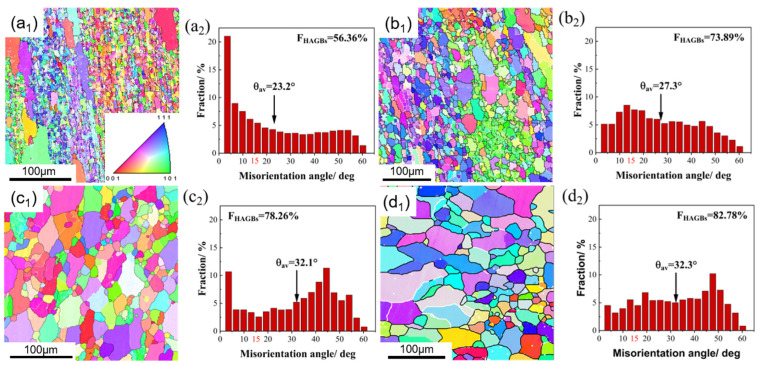
EBSD IPF and misorientation angle graphs of the extruded alloys: (**a_1_**) IPF map of extruded CC alloy; (**a_2_**) misorientation angle distribution of extruded CC alloy; (**b_1_**) IPF map of CC-SS alloy; (**b_2_**) misorientation angle distribution of CC-SS alloy; (**c_1_**) IPF map of extruded ACC alloy; (**c_2_**) misorientation angle distribution of extruded ACC alloy; (**d_1_**) IPF map of ACC-SS alloy; (**d_2_**) misorientation angle distribution of ACC-SS alloy.

**Figure 10 materials-16-03042-f010:**
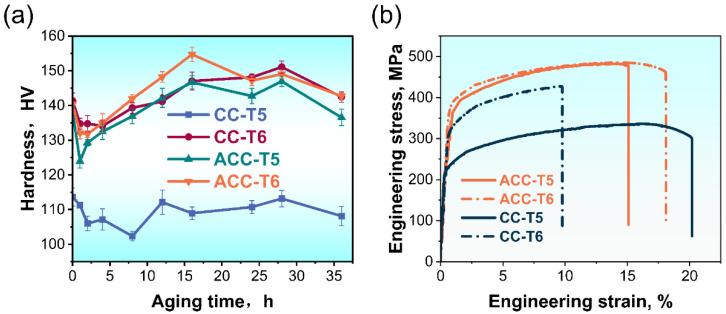
Mechanical properties of 2024 Al alloy under T5 and T6 treatment: (**a**) microhardness; (**b**) engineering stress-strain curves after 16 h AA treatment.

**Figure 11 materials-16-03042-f011:**
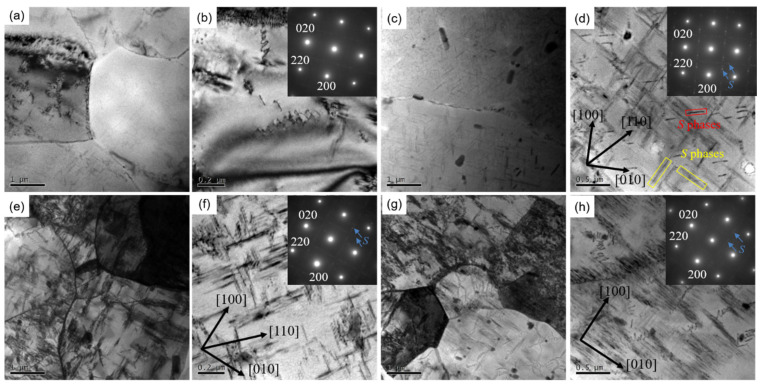
TEM microstructures and SAED patterns of: (**a**,**b**) CC-T5 specimen; (**c**,**d**) CC-T6 specimen; (**e**,**f**) ACC-T5 specimen; (**g**,**h**) ACC-T6 specimen.

**Figure 12 materials-16-03042-f012:**
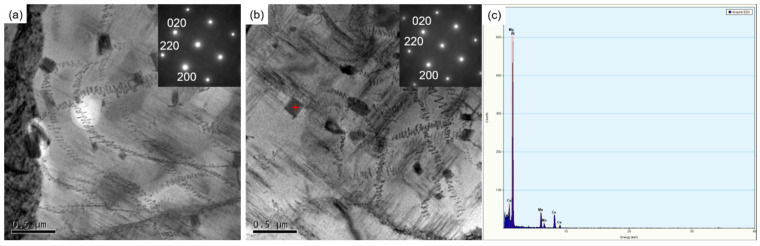
TEM microstructures showing the interaction between dispersoids and dislocations in ACC samples after: (**a**) T5 treatment; (**b**) T6 treatment and (**c**) EDX analysis of dispersoid.

**Figure 13 materials-16-03042-f013:**
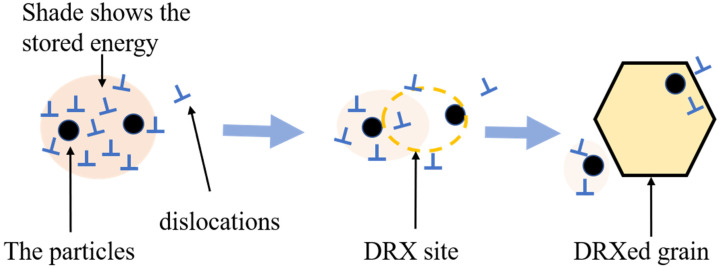
The schematic illustration indicates the effect of the secondary particles on the DRX process.

**Table 1 materials-16-03042-t001:** The values of material constants of CC and ACC 2024 Al alloys.

	α/MPa^−1^	n	*Q*/(KJ·mol^−1^)	A/s^−1^
CC	0.019974	4.0621	224.0402	7.2710 × 10^14^
ACC	0.014391	5.3905	297.8640	3.2845 × 10^20^

**Table 2 materials-16-03042-t002:** The values of mechanical properties of CC and ACC 2024 Al alloys.

Sample	UTS/MPa	YS/MPa	Elongation/%
CC-T5	335	222.9	20.2
ACC-T5	481	361	15.1
CC-T6	427	321.7	9.78
ACC-T6	485	379.2	18.1

## Data Availability

The data presented in this study are available on the request from the corresponding author.

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
