# Peer review of "Influence of Pre-Heat Treatment on the Deformation Behaviors, Microstructural Characteristics, and Mechanical Properties of a Continuously Cast Al-Cu-Mg Alloy during Continuous Extrusion Process"

_materials, 2023, doi:10.3390/ma16083042_

Round 1
Reviewer 1 Report
The authors investigated the effect of a pre-heat treatment on the microstructure and mechanical properties of continuously cast 2024 Al alloys subsequently extruded and heat treated. The investigation was thorough and ambitious, including compression tests, modelling and characterization by LOM, SEM, TEM and EBSD.
I only lack a figure showing a comparison between the experimentally obtained stress-strain curves and the corresponding curves obtained using the fitted expressions.
Reviewer 2 Report
1. In Figure 2(b) could be better to carry out TEM analysis to determine exactly the amount of Cu and Mg which is segregated along the boundaries.
2. XRD studies may be useful to take into the account the phases of the alloys.
3. It would be useful to add a table where UTS, YS, elongation can be shown.
Reviewer 3 Report
The manuscript in its current state could be accepted after major revision.
Before accepting the manuscript in its current state, some aspects should be corrected.
1.- The manuscript presents many abbreviations that need to be described.
2.- Figure 1 is confusing and needs to be clarified.
3.- It is necessary to include how many samples were evaluated for the tensile and hardness tests. The graphs are supposed to show an average hardness value and the representative stress/strain curve. However, the error in the first and the average and the error in the last should be included.
4.- The temperature at which the water quenching was carried out in the different treatments should be specified.
5.- Specify what mean by "the OM characteristic of CC alloy reveals".
6.- In figure 2d, clarify whether the particles observed correspond to particles formed during homogenization (what type of particles are they: precipitates T or other types of second phases) or are remnants of interdendritic segregation that were not dissolved because the saturation was reached in the matrix phase.
7.- In figure 2d, an EDS analysis should be included.
8.- In figure 3, the scale in the y-axis (True stress, MPa) must be the same for the four graphs.
9.- Check for misspellings like the word "higaher" on line 217
10.- Clarify the caption of figure 4.
11.-The x and y axes scale should be the same in figures 4 (e) and (f).
12.- The misorientation angle plots in figure 5 should go with the same scale on the x and y axes.
13.- Clarify why the ACC alloy in finite element analysis has a higher outlet temperature than the CC alloy during the extrusion process.
14.- Clarify why the ACC-T5 sample obtains the same mechanical properties as the CC-T6 and ACC-T6 samples. ¿Is precipitation of second phases generated during the extrusion process?
15.- Correct writing errors such as the one observed in line 354 underlining on S letter
16.- Clarify the type and origin of dispersoids observed in figure 10. Are they T precipitates or remnants of interdendritic segregation?
17.- Add further discussion about the size and spatial orientation of the precipitates in figure 9. The size of the precipitates is different in the samples (c) & (d) CC-T6 specimen; (e) & (f) ACC-T5 specimen; (g) & (h) ACC-T6 specimen. In addition, precipitates oriented at 90°, {001}, are observed; however, for the (d) CC-T6 specimen, other precipitates are observed in a horizontal direction. To which phase do they correspond, and why does their orientation differ?
Round 2
Reviewer 2 Report
You have replied very well all my questions